# Characterization of the Fungitoxic Activity on *Botrytis cinerea* of *N*-phenyl-driman-9-carboxamides

**DOI:** 10.3390/jof7110902

**Published:** 2021-10-26

**Authors:** Ricardo Melo, Verónica Armstrong, Freddy Navarro, Paulo Castro, Leonora Mendoza, Milena Cotoras

**Affiliations:** 1Núcleo de Química y Bioquímica, Facultad de Estudios Interdisciplinarios, Universidad Mayor, Santiago 8580745, Chile; ricardo.melo@mayor.cl; 2Departamento de Química Orgánica, Facultad de Química, Pontificia Universidad Católica de Chile, Santiago 7820436, Chile; aarmstrl@uc.cl; 3Laboratorio de Micología, Facultad de Química y Biología, Universidad de Santiago de Chile, Santiago 7820436, Chile; freddy.navarro@usach.cl (F.N.); paulo.castro@usach.cl (P.C.)

**Keywords:** *Botrytis cinerea*, antifungal activity, *N*-phenyl-driman-9-carboxamides, mode of action, oxidative phosphorylation inhibitors

## Abstract

A total of 12 compounds were synthesized from the natural sesquiterpene (-) drimenol (compounds 4 to 15). The synthesized compounds corresponded to *N*-phenyl-driman-9-carboxamide derivatives, similar to some fungicides that inhibit the electron-transport chain. Their structures were characterized and confirmed by ^1^H NMR, ^13^C NMR spectroscopy, and mass spectrometry. Compounds 5 to 15 corresponded to novel compounds. The effect of the compounds on the mycelial growth of *Botrytis cinerea* was evaluated. Methoxylated and chlorinated compounds in the aromatic ring (compounds 6, 7, 12, and 13) exhibited the highest antifungal activity with IC_50_ values between 0.20 and 0.26 mM. On the other hand, the effect on conidial germination of *B. cinerea* of one methoxylated compound (6) and one chlorinated compound (7) was analyzed, and no inhibition was observed. Additionally, compound 7 decreased 36% the rate of oxygen consumption by germinating conidia.

## 1. Introduction

*Botrytis cinerea* is a facultative phytopathogenic fungus causing grey mold disease in a wide range of hosts. It infects aerial parts of the plants such as leaves, stems, fruits, and flowers of 586 genera of vascular plants, representing over 1400 ornamental and agriculturally important plant species. [1]. Due to the economic losses caused by this pathogen and the difficulties in its control, *B. cinerea* infection represents one of the main phytosanitary problems worldwide [2,3].

The use of synthetic fungicides is the main strategy to control this pathogen. The mechanisms of action of these fungicides are as diverse as their structures [4]. Succinate dehydrogenase inhibitors (SDHI), belonging to the carboxamide family, are the most used fungicides to control *B. cinerea* [5]. These compounds can bind to the union site of the ubiquinone in the Complex II of the mitochondrial electron chain transport through hydrophobic interactions among the aromatic ring of the fungicide and a proline residue and two tryptophan residues of the Complex II. On the other hand, a hydrogen bond is produced among the carbonyl group of the fungicide and a tyrosine residue of Complex II. These interactions interrupt electronic transport from [3Fe–4S] cluster to ubiquinone in Complex II, affecting the cellular respiration in *B. cinerea* [5,6,7].

Resistance of *B. cinerea* to fungicides is a permanent problem [8,9,10,11], so various strategies have been developed, such as increasing the doses of fungicides and the use of alternative fungicides [12,13,14]. Secondary metabolites are an option for searching for new fungicides since some of these compounds or their derivatives have exhibited fungitoxic activity against *B. cinerea* [15,16,17,18], which shows a strong relationship between some functional groups in its structure with fungitoxic activity [19,20,21,22,23]. Drimane sesquiterpenes are secondary metabolites with bicyclic skeletons that present numerous biological activities [24]. It has been reported that drimenol, obtained from Canelaceas (Canellaceae) species, showed cytotoxic, antibacterial, antifungal, and anti-inflammatory activity. [24]. It was also determined that drimenal has a higher fungitoxic activity than drimenol, both extracted from the bark of winter cinnamon (*Drimys winteri* Forst.), with IC_50_ of 81.8 μg/mL (0.37 mmol/L) and >125 μg/mL (>0.6 mmol/L), respectively [25]. The increase in fungitoxic activity observed with drimenal is not observed when drimenol is acetylated, while the epoxy acetate derivative shows a decrease in fungitoxic activity concerning drimenol [26].

This study proposed to analyze the antifungal activity against *B. cinerea* of 12 SDHI-like compounds synthesized from drimenol by reaction between this compound and substituted anilines. Furthermore, we analyzed the possible mode of action of two of the most active compounds on *B. cinerea*.

## 2. Materials and Methods

### 2.1. General Conditions

Reagents used in synthesis were obtained from Sigma-Aldrich Co. (St. Louis, MO, USA). Organic solvents were obtained from Merck Química Chilena (Santiago, Chile). Column chromatography was carried out using Gel 60 (0.063–0.200 mm) (Merck, Darmstadt, Germany), and thin-layer chromatography (TLC) was performed on Merck Kiesegel 60 F254 0.2 cm.

Synthesis of the compounds was carried out from natural (-)drimenol extracted from the bark of winter cinnamon [27], provided by the Natural Product Laboratory of the Universidad Católica de Chile.

A Thermo Scientific gas chromatography coupled with a mass spectrometer (GC-MS) system (GC model: Trace 1300 and MS model: TSQ8000Evo) (Waltham, MA, USA) was used to analyze the samples. The separation was performed on a 60 mm × 0.25 mm internal diameter fused silica capillary column coated with 0.25 µm film Rtx-5MS. The oven temperature was maintained at 40 °C for 5 min, and then set from 40 to 80 °C at 5 °C/min, then from 80 to 300 °C at 30 °C/min, and finally maintained at 300 °C for 25 min. A splitless injection was used with helium as a carrier gas, and the flow rate was 1.2 mL/min. Mass spectra were recorded over a range of 40 to 400 atomic mass units at 0.2 s/scan, the solvent cut time was 11 min, and the ionization energy was 70 eV.

### 2.2. Synthesis of Compounds

#### 2.2.1. Synthesis of Driman-11-ol (2)

Palladium/carbon (10%) was added to a solution of drimenol (1) (2.25 mmol) dissolved in methanol p.a. (50 mL). This mixture was treated with molecular hydrogen for 4 h, and the mixture was filtered out. The solvent was concentrated under vacuum to afford crude reaction, and then it was purified through column chromatography using hexane:acetate 8:2 as mobile phase. The NMR signals (see Appendix B) were compared with previous reports [28,29,30].

#### 2.2.2. Synthesis of Driman-11-oic Acid (3)

Jones reagent (CrO_3_/H_2_SO_4_) drops were added to a solution of (2) (2.23 mmol) in acetone p.a (50 mL) until the obtention of a permanent orange coloration in the solution. The mixture was concentrated in a vacuum to obtain a green crude which was then extracted with ethyl acetate. This extract was concentrated in a vacuum, and an acid-base extraction was carried out. The organic phase was evaporated to dryness, obtaining driman-11-oic acid (3) as a white-colored solid. The NMR signals analysis (see Appendix B) showed that these signals correlate with previous reports [31].

#### 2.2.3. Synthesis of Drimancarboxamide Derivatives

Compound 3 (0.42 mmol) was dissolved in the minimum amount of thionyl chloride (SOCl_2_) and three drops of *N*,*N*-dimethylformamide (DMF). The solution was refluxed for 3 h, then was distilled at reduced pressure and dissolved in dichloromethane (CH_2_Cl_2_). 0.42 mmol of the corresponding aniline (1:1 to chloride compound synthesized from (3)) and 0.42 mmol of triethylamine in CH_2_Cl_2_ anhydrous were added drop by drop. The mixture remained at 0 °C. The product of the reaction was purified through column chromatography. Anilines used were: aniline, 3,4-dichloroaniline, 3,4-dimethoxyaniline, 3,4-dimethylalanine, 3,5-dichloroaniline, 3,5-dimethoxyaniline, 3-chloroaniline, 4-chloraniline, 3-methylaniline, 4-methylaniline, 3-methoxyaniline, and 4-methoxyaniline. Thus, 12 compounds were obtained (compounds 4 to 15).

#### 2.2.4. Spectroscopic Data of Compounds 4–15

*2,5,5,8a-tetramethyl-N-phenyldecahydronaphthalene-1-carboxamide* or *N-phenyl-driman-9-carboxamide* (4); light yellow solid, yield 40%, and m.p.:194 °C. ^1^H NMR (400 MHz, CDCl_3_) δ 7.43 (2H, d, *J* = 8.1 Hz, Ha`), 7.20 (2H, t, *J* = 7.7 Hz, Hb`), 6.99 (1H, t, *J* = 6.9 Hz, Hc`), 1.07 (3H, s, CH_3_), 0.81 (3H, d, *J* = 6.5 Hz, CH_3_), 0.79 (3H, s, CH_3_), 0.77 (3H, s, CH_3_). ^13^C NMR (100 MHz, CDCl_3_) δ 171.67 (C=O), 137.17 (C1`), 127.86 (C3`), 122.88 (C4`), 118.84 (C2`), 61.10 (C1), 55.45 (C4a), 41.17 (C6), 38.93 (C8), 37.30 (C8a), 33.44 (C3), 32.44 (CH_3_–C2), 32.20 (C5), 31.55 (CH_3_–C5), 20.54 (C2), 17.12 (C4), 16.41 (C7), 16.11 (CH_3_–C8a), 15.11 (CH_3_–C5). GC-MS RI_(Rtx-5ms)_ = 2809.6, C_21_H_31_NO EI-MS *m*/*z*: 93 (100); 123 (31.14); 135 (28.12); 205 (46.93); 220 (27.3); [M]^+^ = 313 (26.63).

*2,5,5,8a-tetramethyl-N-(3,4-dimethylphenyl)decahydronaphthalene-1-carboxamide* or *N-(3,4-dimethylphenyl)-driman-9-carboxamide* (5); purple solid, yield 37%, and m.p.:157 °C. ^1^H NMR (400 MHz, CDCl_3_) δ 7.27 (1H, d, *J* = 2.1 Hz, Ha`), 7.12 (1H, dd, *J* = 8.1, 2.1 Hz, Hc`), 6.94 (1H, d, *J* = 8.1 Hz, Hb`),2.12 (6H, s, CH_3_ (C3`) CH_3_ (C4`)), 1.07 (3H, s, CH_3_), 0.8 (3H, d, *J* = 6.5 Hz, CH_3_), 0.79 (3H, s, CH_3_), 0.77 (3H, s, CH_3_). ^13^C NMR (100 MHz, CDCl_3_) δ171.11 (C=O), 136.07 (C3′), 134.56 (C1′), 131.32 (C4′), 128.80 (C5′), 120.37 (C2′), 116.46 (C6′), 66.80 (C1), 53.64 (C4a), 41.07 (C6), 39.61 (C8), 36.80 (C8a),34.57 (C3), 32.58 (CH_3_–C2), 32.22 (C5), 29.00 (C2), 20.79 (CH_3_-C5), 20.59, 19.92 (CH_3_–C8a), 18.81 (CH_3_–C3′), 18.13, 17.70 (CH_3_–C4′), 13.51 (CH_3_–C5). GC-MS RI_(Rtx-5ms)_ = 2971.5, C_23_H_33_NO EI-MS *m*/*z*: 121 (100); 122 (17.10); 123 (12.93); 163 (21.16); [M]^+^ = 341 (20.21).

*2,5,5,8a-tetramethyl-N-(3,4-dimethoxyphenyl)decahydronaphthalene-1-carboxamide* or *N-(3,4-dimethoxyphenyl)-driman-9-carboxamide* (6); brown oil, and 40% yield. ^1^H NMR (200 MHz, CDCl_3_) δ7.41 (1H, d, *J* = 1.8 Hz, Ha`), 6.86 (1H, dd, *J* = 8.2, 1.7 Hz, Hc`), 6.75 (1H, d, *J* = 8.2 Hz, Hb`), 3.83 (6H, s, OMe (C3`), OMe (C4`)), 1.14 (3H, s, CH_3_), 0.90 (3H, s, CH_3_), 0.86 (3H, s, CH_3_), 0.85 (3H, s, CH_3_). ^13^C NMR (100 MHz, CDCl_3_) δ 172.34 (C=O), 149.01 (C3`), 145.61 (C4`), 131.80 (C1`), 111.65 (C6`), 111.31 (C5`), 104.86 (C2`), 67.90 (C9), 56.21 (OMe–C4`), 55.90 (OMe–C3`), 54.74 (C4a), 42.16 (C6), 40.71 (C8), 37.91 (C8a), 35.68 (C3), 33.70 (C5), 33.33 (CH_3_-C2), 30.12 (C2), 21.90 (CH_3_–C5), 21.68 (CH_3_–C8a), 21.07 (C4), 18.81 (C7), 14.64 (CH_3_–C5). GC-MS RI_(Rtx-5ms)_ = 3137.7, C_23_H_35_NO_3_ EI-MS *m*/*z*: 69 (10.6); 138 (16.96); 153 (100); 154 (10.01); [M]^+^ = 373 (16.48).

*2,5,5,8a-tetramethyl-N-(3,4-dichlorophenyl)decahydronaphthalene-1-carboxamide* or *N-(3,4-dichlorophenyl)-driman-9-carboxamide* (7); orange oil, and 40% yield. ^1^H NMR (400 MHz, CDCl_3_) δ 7.75 (1H, s, Ha`), 7.31 (2H, d, *J* = 1.1 Hz, Hb` y Hc`), 1.12 (3H, s, CH_3_), 0.86 (3H, s, CH_3_), 0.85 (3H, s, CH_3_), 0.84 (3H, s, CH_3_). ^13^C NMR (101 MHz, CDCl_3_) δ 172.73 (C11), 137.32 (C1`), 132.80 (C3`), 130.49 (C5`), 127.36 (C4`), 121.84 (C2`), 119.36 (C6`), 68.07 (C9), 54.80 (C5), 42.16 (C6), 40.88 (C1), 38.10 (C10), 35.64 (C7), 33.71 (C4), 33.39 (C13), 30.21 (C8), 21.92 (C14), 21.69 (C12), 21.03 (C6), 18.82 (C2), 14.65 (C15). GC-MS RI_(Rtx-5ms)_ = 3162.1, C_21_H_29_Cl_2_NO EI-MS *m*/*z*: 69 (43.47); 97 (43.23); 123 (56.37); 137 (46.52); 193 (100); [M]^+^ = 381 (10.56); [M+2]^+^ = 383 (7.36); [M+4]^+^ = 385 (1.27).

*2,5,5,8a-tetramethyl-N-(3,5-dimethoxyphenyl)decahydronaphthalene-1-carboxamide* or *N-(3,5-dimethoxyphenyl)-driman-9-carboxamide* (8); orange solid, 30% yield, and m.p.: 121°C. ^1^H NMR (400 MHz, CDCl_3_) δ 6.78 (d, *J* = 1.7 Hz, 2H, Ha`), 6.20 (t, *J* = 2.2 Hz, 1H, Hb`), 3.76 (s, 6H, 2 OMe), 1.10 (d, *J* = 7.4 Hz, 3H, CH_3_), 0.86 (s, 6H, 2CH_3_), 0.84 (s, 3H, CH_3_). ^13^C NMR (101 MHz, CDCl_3_) δ 179.88 (C11), 161.41(C3′, C5′), 140.53 (C1′), 98.50 (C2′), 98.30 (C6′), 96.91 (C4’), 59.66 (C1), 56.89 (CH_3_–C3′), 56.29 (CH_3_–C5′), 55.77 (C4a), 42.51 (C6), 39.92 (C8a), 38.74 (C8), 37.63 (C3), 34.38 (C2), 33.56 (C5), 32.99 (CH_3_–C5), 31.91 (CH_3_–C5), 21.97 (C4), 21.97 (CH_3_–C2), 18.52 (C7), 16.48 (CH_3_–C8a). GC-MS RI_(Rtx-5ms)_ = 3221.3, C_23_H_35_NO_3_ EI-MS *m*/*z*: 69.08(10.54); 153 (100); 154 (15.54); 195 (12.39); [M]^+^ = 373 (11.57).

*2,5,5,8a-tetramethyl-N-(3,5-dichlorophenyl)decahydronaphthalene-1-carboxamide* or *N-(3,5-dichlorophenyl)-driman-9-carboxamide* (9); white solid, yield 33%, and m.p.:180 °C. ^1^H NMR (400 MHz, CDCl_3_) δ 7.46 (d, *J* = 1.1 Hz, 2H, Ha`), 7.02 (d, *J* = 1.4 Hz, 1H, Hb`), 1.32 (s, 3H, CH_3_), 1.20 (d, *J* = 7.5 Hz, 3H, CH_3_), 0.85 (s, 3H, CH_3_), 0.84 (s, 3H, CH_3_). ^13^C NMR (101 MHz, CDCl_3_) δ 173.59 (C=O), 140.41, 135.46, 124.16, 118.57, 62.45, 56.84, 42.56, 40.38, 38.76, 34.77, 33.82, 33.61, 32.93, 21.95, 18.51, 17.80, 17.52, 16.57. GC-MS RI_(Rtx-5ms)_ = 3044.8, C_21_H_29_Cl_2_NO EI-MS *m*/*z*: 49 (22.33); 69 (68.07); 97 (83.77); 123 (91.99); 193 (100); [M]^+^ = 381 (16.29); [M+2]^+^ = 383 (10.53); [M+4]^+^ = 385 (1.87).

*2,5,5,8a-tetramethyl-N-(4-methylphenyl)decahydronaphthalene-1-carboxamide* or *N-(4-methylphenyl)-driman-9-carboxamide* (10); white solid, yield 30%, and m.p.:176 °C. ^1^H NMR (400 MHz, CDCl_3_) δ 7.41–7.35 (m, 2H, Ha`), 7.14 (s, 1H, NH), 7.09 (d, *J* = 8.1 Hz, 2H, Hb`), 2.30 (s, 3H, Ar–CH_3_), 1.35 (s, 3H, CH_3_), 1.22 (d, *J* = 7.4 Hz, 3H, CH_3_), 0.86 (s, 3H, CH_3_), 0.86 (s, 3H, CH_3_). ^13^C NMR (101 MHz, CDCl_3_) δ 172.46 (C=O), 135.65 (C1′), 133.45 (C4′), 129.36 (C3′C5′), 119.97 (C2′C6′), 68.00 (C1), 62.14 (C4a), 56.52 (C6), 54.74 (C8a), 42.25 (C8), 39.99 (C3), 38.35 (C2), 34.51 (C5), 33.24 (CH_3_–C5), 32.62 (CH_3_–C8a), 21.58 (CH_3_–C4′), 20.82 (C4), 18.18 (CH_3_–C2), 17.47 (C7), 16.14 (CH_3_–C5). GC-MS RI_(Rtx-5ms)_ = 2909.6, C_22_H_33_NO EI-MS *m*/*z*: 69 (9.43); 106 (8.98); 107 (100); 108 (9.31); [M]^+^ = 327 (5.95).

*2,5,5,8a-tetramethyl-N-(3-methylphenyl)decahydronaphthalene-1-carboxamide* or *N-(3-methylphenyl)-driman-9-carboxamide* (11); white solid, yield 30%, and m.p.:173 °C. ^1^H NMR (400 MHz, CDCl_3_) δ 7.43 (d, *J* = 7.0 Hz, 1H, Hd`), 7.24–7.12 (m, 3H, Hc´, Ha`, NH), 6.89 (t, *J* = 5.9 Hz, 1H, Hb`), 2.32 (s, 3H, Ar–CH_3_), 1.36 (s, 3H, CH_3_), 1.23 (d, *J* = 7.4 Hz, 3H, CH_3_), 0.87 (s, 3H, CH_3_), 0.86 (s, 3H, CH_3_). ^13^C NMR (101 MHz, CDCl_3_) δ 172.62 (C=O), 138.86 (C3`), 138.15 (C1`), 128.69 (C5`), 124.69 (C4`), 120.60 (C2`), 116.94 (C6`), 68.15 (C1), 62.25 (C4a), 56.53 (C6), 42.24 (C8a), 40.02 (C8), 38.36 (C3), 34.52 (C2), 33.47(CH_3_–C2), 32.63(CH_3_–C5), 21.58 (CH_3_–C3`), 18.18 (C4), 17.47 (CH_3_–C8a), 17.14 (C7), 16.14 (CH_3_–C5). GC-MS RI_(Rtx-5ms)_ = 2882.4, C_22_H_33_NO EI-MS *m*/*z*: 69 (11.32); 107 (100); 108 (12.27); 123 (10.59); 149 (14.12); [M]^+^ = 327 (12.25).

*2,5,5,8a-tetramethyl-N-(4-methoxyphenyl)decahydronaphthalene-1-carboxamide* or *N-(4-methoxyphenyl)-driman-9-carboxamide* (12); white solid, yield 37%, and m.p.:124 °C. ^1^H NMR (400 MHz, CDCl_3_) δ 7.32 (d, *J* = 8.3 Hz, 2H, Ha`), 6.74 (d, *J* = 8.4 Hz, 2H, Hb`), 3.69 (s, 3H, Ar-O-Me), 1.27 (s, 3H, CH_3_), 1.15 (d, *J* = 7.4 Hz, 3H, CH_3_), 0.78 (s, 6H, 2CH_3_); ^13^C NMR (101 MHz, CDCl_3_) δ 172.57 (C=O), 156.12 (C4`), 131.40 (C1`), 121.85 (C2`), 114.03 (C3`), 61.86 (C1), 56.50 (OMe–C3′), 55.49 (C4a), 42.24 (C6), 39.97 (C8a), 38.30 (C8), 34.52 (C3), 33.47 (C5), 33.23 (C2), 32.61 (CH_3_–C2), 21.58 (CH_3_–C5), 18.19 (C4), 17.47 (CH_3_–C8a), 17.17 (C7), 16.16 (CH_3_–C5). GC-MS RI_(Rtx-5ms)_ = 3060.4, C_22_H_33_NO_2_ EI-MS *m*/*z*: 69 (9.10); 108 (10.26); 123 (100); 124 (12.09); [M]^+^ = 343 (10.47).

*2,5,5,8a-tetramethyl-N-(3-methoxyphenyl)decahydronaphthalene-1-carboxamide* or *N-(3-methoxyphenyl)-driman-9-carboxamide* (13); white solid, yield 37%, and m.p.:127 °C. ^1^H NMR (400 MHz, CDCl_3_) δ 7.31 (s, 1H, NH), 7.20–7.13 (m, 2H, Ha`, Hc`), 6.94 (d, *J* = 7.9 Hz, 1H, Hb`), 6.63 (d, *J* = 7.4 Hz, 1H, Hd`), 3.79 (s, 3H, Ar–O–CH_3_), 1.35 (s, 3H, CH_3_), 1.22 (d, *J* = 7.4 Hz, 3H, CH_3_), 0.86 (s, 6H, 2CH_3_). ^13^C NMR (101 MHz, CDCl_3_) δ 178.61 (C=O), 172.65 (C3`), 139.53 (C1`), 129.54 (C5`), 111.82 (C6`), 109.92 (C4`), 105.32 (C2`), 64.51 (C1), 56.52 (OMe–C3′), 55.32 (C4a), 42.23 (C6), 40.00 (C8a), 38.36 (C8), 34.49 (C3), 33.47 (C5), 32.60 (C2), 21.58 (CH_3_–C2), 18.17 (CH_3_–C5), 17.46 (C4), 17.14 (CH_3_–C8a), 16.16 (C7), 14.48 (CH_3_–C5). GC-MS RI_(Rtx-5ms)_ = 2957.6, C_22_H_33_NO_2_ EI-MS *m*/*z*: 69 (10.58); 123 (100); 124 (11.32); 165 (9.50) [M]^+^ = 343 (10.27).

*2,5,5,8a-tetramethyl-N-(4-chlorophenyl)decahydronaphthalene-1-carboxamide* or *N-(4-chlorophenyl)-driman-9-carboxamide* (14), white solid, yield 35%, and m.p.:181 °C. ^1^H NMR (400 MHz, CDCl_3_) δ 7.45 (d, *J* = 8.1 Hz, 2H, Ha`), 7.25 (d, *J* = 8.5 Hz, 2H, Hb`), 1.97 (d, *J* = 4.1 Hz, 1H, H-C9), 1.34 (s, 3H, CH_3_), 1.21 (d, *J* = 8.0 Hz, 3H, CH_3_), 0.86 (s, 3H, CH_3_), 0.86 (s, 3H, CH_3_). ^13^C NMR (101 MHz, CDCl_3_) δ 172.63(C=O), 136.73 (C1`), 128.88 (C3`C4`), 121.09 (C2`), 62.25 (C1), 56.53 (C4a), 42.20 (C6), 40.05 (C8a), 38.38 (C8), 34.46 (C3), 33.46 (C5), 33.24 (C2), 32.59 (CH_3_–C2), 21.56 (CH_3_–C5), 18.15 (CH_3_–C8a), 17.43 (C4), 17.09 (C7), 16.12 (CH_3_–C5). GC-MS RI_(Rtx-5ms)_ = 3024.5, C_21_H_30_ClNO EI-MS *m*/*z*: 69 (59.81); 81.1 (46.52); 97 (56.88); 109 (44.15); 123 (60.75); 127 (100); 137 (47.33); 193 (50.09); 205 (64.29); 220 (70.66); [M]^+^ = 347 (17.83); [M+2]^+^ = 349 (6.13).

*2,5,5,8a-tetramethyl-N-(3-chlorophenyl)decahydronaphthalene-1-carboxamide* or *N-(3-chlorophenyl)-driman-9-carboxamide* (15); white solid, yield 35%, and m.p.:204 °C. ^1^H NMR (400 MHz, CDCl_3_) δ 7.66 (s, 1H, NH), 7.29 (d, *J* = 8.3 Hz, 1H, Hd`), 7.20 (m, 2H, Ha`, Hc`), 7.04 (d, *J* = 7.9 Hz, 1H, Hb`), 1.98 (d, *J* = 4.2 Hz, 1H, H-C9), 1.34 (s, 3H, CH_3_), 1.21 (d, *J* = 7.4 Hz, 3H, CH_3_), 0.86 (s, 3H, CH_3_), 0.85 (s, 3H, CH_3_). ^13^C MR (101 MHz, CDCl_3_) δ 172.73 (C=O), 137.32 (C1`), 132.80 (C3`), 130.49 (C5`), 127.36 (C4`), 121.84 (C2`), 119.36 (C6`), 68.07 (C1), 54.80 (C4a), 42.16 (C4), 40.88 (C8), 38.10 (C8a), 35.64 (C3), 33.71 (C5), 33.39 (CH_3_–C2), 30.21 (C2), 21.92 (CH_3_–C5), 21.69 (CH_3_–C8a), 21.03 (C4), 18.82 (C7), 14.65 (CH_3_–C5). GC-MS RI_(Rtx-5ms)_ = 3005.3, C_21_H_30_ClNO EI-MS *m*/*z*: 69 (82.53); 81 (63.65); 97 (83.41); 109 (66.19); 123 (100); 137 (71.73); 193 (75.81); 205 (82.95); 220 (87.76); [M]^+^ = 347 (36.53); [M+2]^+^ = 349 (12.26).

All spectra are accompanied in Appendix A.

### 2.3. Fungal Isolate and Culture Conditions

*Botrytis cinerea* isolate G29 originally isolated from naturally infected grapes (*Vitis vinifera*) [32] was used in this study. The fungus was grown in the dark on malt-yeast extract agar medium (2% (*w*/*v*) malt extract, 0.2% (*w*/*v*) yeast extract and 1.5% (*w*/*v*) agar), on soft agar medium (2% (*w*/*v*) malt extract, 0.2% (*w*/*v*) yeast extract, and 0.6% (*w*/*v*) agar) or in liquid minimum medium (1 g L^−1^ KH_2_PO_4_, 0.5 g L^−1^ K_2_HPO_4_, 0.5 g L^−1^ MgSO_4_ × 7H_2_O, 0.5 g L^−1^ KCl, 0.001 g L^−1^ FeSO_4_ × 7H_2_O, and 4.6 g L^−1^ ammonium tartrate) The culture media was adjusted to pH 6.0 and supplemented with 1% glucose (Merck Millipore, Darmstadt, Germany).

### 2.4. Determination of the Antifungal Activity of Compounds against B. cinerea

The effect of the compounds on the mycelial growth of *B. cinerea* was determined as described by Mendoza et al. (2015) [22]. Compounds dissolved in acetone at final concentrations of 0.09, 0.18, 0.36 mmol/L for drimenol, and 0.06, 0.12, 0.24 mmol/L for compounds 4–15 were added to Petri dishes containing malt-yeast extract agar medium. The final acetone concentration was identical in the control and treatment assays. After acetone evaporation in a laminar flow cabinet, the culture medium was inoculated with 0.5 cm agar disks from an actively growing culture of *B. cinerea*. Cultures were incubated in the dark at 22 °C. Mycelium diameter was measured daily, and results were expressed as IC_50_, determined by the inhibition of radial growth against compound concentrations using Probit analysis. Each experiment was done at least in triplicate.

Also, the effect of synthesized compounds on conidia germination was determined. Conidia germination assays were carried out on microscope slides coated with soft agar medium (2 mm thickness). Culture media were inoculated with dry conidia obtained from sporulated mycelia. Slides were placed in a humid chamber (90% relative humidity) and incubated in the dark at 22 °C. The number of germinated conidia was counted at 1-h intervals through microscope observation. The germination percentage was calculated by counting germinated conidia in five microscope fields containing approximately one hundred conidia. Conidia with a germinative tube length similar to conidial diameter were considered germinated conidia, according to the protocol described by Mendoza et al. (2015) [22]. Each assay was performed at least in triplicate.

### 2.5. Evaluation of Possible Mode of Action on B. cinerea of the Most Active Synthesized Compounds

The effects of one of the most active compounds on oxidative phosphorylation and oxygen consumption by germinating conidia in *B. cinerea* were evaluated.

#### 2.5.1. Effect of Salicylhydroxamic Acid (SHAM) on Mycelial Growth of *B. cinerea*

The effect of SHAM, an inhibitor of the cyanide-resistant respiration pathway in fungal mitochondria [33], at 5 mM or compounds 6 and 7 at 0.29 mM plus 5 mM SHAM was evaluated on the mycelial growth of *B. cinerea* as described in 2.4. Each assay was performed at least in triplicate.

#### 2.5.2. Effect of Compounds on Oxygen Consumption

The commercial kit MitoXpress^®^-Xtra was used as described by Diepart et al. (2010) [34] to quantify the oxygen consumption with some modifications. In a 96-well plate, 1 × 10^5^ conidia/mL were inoculated in 200 µL of Minimum Medium, and the culture was incubated at 22 °C overnight.

After the incubation, the culture medium was removed, and 150 µL of fresh Minimum Medium was added. Then, 10 µL of culture medium was added to “solvent control” wells, 10 µL of antimycin A (100 µM) to “AA control” wells, and 10 µL of compound 6 or 7 at 0.29 mM to “treatment” wells. In the “GO control” wells, 10 µL of glucose oxidase (1 mg/mL) were added to wells containing 150 µL of fresh Minimum Medium without conidia. Later, 10 µL of MitoXpress^®^-Xtra was added to each well, and the wells were covered immediately with 100 µL of mineral oil. Fluorescence intensity was measured at 22 °C in a multimodal-microplate reader, model Synergy HT (BioTek Instruments, Winooski, VT, EE.UU) using standard 340 nm excitation and 645 nm emission filters. Fluorescence intensity was measured every 2 min for 120 min. The rate of oxygen consumption was determined from the slope of fluorescence in the time through linear regression. The values were corrected concerning the blank and normalized with the initial intensity.

## 3. Results and Discussion

### 3.1. Synthesis and Structural Determination

A total of 12 new drimancarboxamides were synthesized through a four-step route (Figure 1). The synthesized compounds are shown in Figure 2, which contain both an aromatic and a sesquiterpenoid part. Compounds 5 to 15 corresponded to novel compounds. All compounds were obtained in yields between 30 and 40%.

The structural determination of reaction intermediaries 2 and 3 was performed by comparing ^1^H NMR and ^13^C NMR spectra with previous reports [28,29,30,31]. The analysis of compounds 4–15 was centered on the ^1^H NMR and ^13^C NMR signals of the aromatic ring and the GC-MS data because the signals of the decalinic ring did not vary (see Section 2.2.4). When using compound 4 as a reference, two triplets (δ 7.20 and δ 6.99) and one doublet (δ 7.43) were observed in the aromatic zone of the ^1^H NMR spectrum. In the compounds substituted in R_1_ and R_2_, the signal pattern changed to one singlet and two doublets, whereas in the compounds substituted in R_1_ and R_3_, only three singlets were observed, while in the compounds substituted in R_2,_ two doublets appeared, and in the compounds substituted in R_1_ one singlet, two doublets, and a multiplet. In addition, in compounds with methyl or methoxy groups, the number of carbon atoms in the molecule can be seen in the ^13^C NMR spectrum.

Signal assignments and compound structures proposal was made from the analysis of two-dimensional homonuclear (COSY) and heteronuclear (HSQC and HMBC) (Section 2.2). Finally, the GC-MS analysis identified the molecular ion of each compound, which led to the proposal of the structure presented in Figure 2.

### 3.2. Determination of the Antifungal Activity of Compounds against B. cinerea

#### 3.2.1. The Effect on Mycelial Growth

All synthesized compounds showed antifungal activity against *B. cinerea* in solid media, which increased when the aromatic ring was substituted (Table 1). However, drimenol only slowed the growth of *B. cinerea* under the conditions tested, similar to observed by Scher and colleagues [25].

Data represent the mean of three different experiments ± standard deviation.

The effect of substitutions in the aromatic ring on biological activity has been widely described, and it is associated with the donor or attractor capacity of electrons [35,36,37]. The synthesized compounds with substitutions of methoxyl and chloride groups in R_2_ and R_3_ positions exhibited the highest fungitoxic activity on *B. cinerea* (6, 7, 12, and 13). The methoxyl group in *ortho* and *para* positions is a strong activator of the ring due to its contribution to electronic density, while the methyl group is a weak activator, and the antifungal activity was lower than methoxyl. In chlorine as a substituent, the inductive effect predominates, behaving as an electro-attractor.

The methoxylated compounds with substitutions in R_2_ and/or R_3_ (6, 12, and 13) exhibited similar IC_50_ values, but methoxyl in R_1_ and R_3_ (compound 8) showed less antifungal activity. In the case of chlorinated compounds (7, 9, 14, and 15), the number and position of the substitution appear to be necessary for their fungitoxic activity, compound 7 is the most active, with substitutions on positions R_2_ and R_3_ of the aromatic ring, while compound 9 (chlorine in R_1_ and R_3_) has the lowest activity among the chlorinated compounds. In both cases, substituent in position R_1_ decreased the activity, while with the substituent in a position *para* to amide, the highest fungitoxic activity was achieved (compound 7), except for compound 13 (methoxyl in R_3_).

The results suggest that the electro-attractor role of chlorine and the possibility of forming hydrogen bonds of the oxygen from the methoxyl group of the compounds could be important properties for the fungitoxic activity against *B. cinerea*. However, the synthesis of more compounds is required to establish the relationship between the position and amount of chlorines and methoxyl groups. Besides, studies about the effect of aromatic ring substituents on the properties of molecules do not consider only the electronic aspects but also steric factors and the possibility to interact through the formation of hydrogen bonds with amino acid residues in the active site of the enzymes [38].

The presence of aromatic rings with substitutions of chlorines is a common characteristic in some fungicides. The diverse action mechanisms of these fungicides have been described as damage of cytoskeleton (fluopicolide) or alteration in the cellular respiration (pyribencarb, boscalid) [39,40,41].

Figure 3 shows a comparison of the effect of the most active synthesized compounds (6 and 7) and the precursor drimenol on the mycelial growth of *B. cinerea*. During the first three days, drimenol produced a similar effect to compounds 6 and 7. Then, the inhibitory effect of drimenol decreased, reaching a growth maximum similar to the control, which suggests that the fungus could detoxify itself from the compound. It has been described that this fungus can detoxify itself using degradative (biotransformation) and non-degradative mechanisms [19,20,21,42,43]. On the other hand, on the 10th day of incubation in the presence of compounds 6 and 7, *B. cinerea* did not reach maximum growth. It is probably more difficult for *B. cinerea* to detoxify itself from these kinds of compounds.

#### 3.2.2. The Effect on Conidia Germination

The effect of compounds on conidial germination was evaluated. Compounds 6 and 7 at 0.15 mM and 0.29 mM did not inhibit conidial germination (Melo, Ricardo. Universidad Mayor, Santiago, Chile. The experiments were carried out as indicated in Section 2.4, but no differences were observed in germination between the presence of the synthesized compounds concerning the control, 2018). Moreover, deformations or morphologic alterations in germinative tubes were not observed as it has previously been reported with other compounds [44].

### 3.3. Evaluation of Possible Mode of Action on B. cinerea

#### 3.3.1. Effect of SHAM on Mycelial Growth of *B. cinerea*

The effect of compounds 6 and 7 as possible inhibitors of the mitochondrial electron transport was analyzed. For this, the mycelial growth in the presence of compounds 6 or 7, and 5 mM SHAM, an inhibitor of the alternative oxidase (AOX) [33], was evaluated (Figure 4). When the principal route of electron transport in oxidative phosphorylation is inhibited, reactive oxygen species (ROS) are produced, which activate AOX. This enzyme acts as a bypass so the oxygen can receive the electrons, recovering the capacity to produce ATP by fungal cells [45,46,47].

SHAM inhibited the *B. cinerea* mycelial growth. When SHAM and compounds 6 or 7 acted together, the inhibition of mycelial growth was higher. After seven days of incubation, the fungal growth was 25% in relation to the control.

There are several cases where a similar behavior was observed and in which the electron-transport chain was affected in *B. cinerea*. When the fungus was incubated with 4,4-dimethylanthracene-1,9,10(4*H*)-trione, which affects the electron-transport chain in the presence of 5 mM SHAM, an inhibition of mycelial growth of 95% was observed [43]. The antifungal activity of 2-allylphenol, a phenolic compound obtained from the fruit-like structure of *Ginkgo biloba* L., increased when 10 mg/mL of SHAM was added to the media [48]. The behavior is also similar when azoxystrobin and antimycin A, well-known inhibitors of mitochondrial electronic transport, are used [49]. The decrease of the mycelial growth when the fungus was treated with the synthesized compounds indicates that the compounds would affect the mitochondrial electron-transport chain.

#### 3.3.2. Effect of Compounds on Oxygen Consumption

The effect of compound 7 on oxygen consumption of germinating conidia was also evaluated. The fluorescence signal of MitoXpress^®^-Xtra is inversely proportional to the oxygen concentration in the suspension, so the increment in oxygen consumption by germinating conidia increases the fluorescence [50]. Figure 5 shows relative fluorescence intensity in the time of germinating conidia suspensions treated with compound 7 and the controls.

The curve slope corresponding to treatment with compound 7 at 0.20 mM shows that the oxygen consumption velocity was 36% lesser than control, while antimycin A reduced oxygen consumption velocity by 85%. This effect indicates that compound 7 would inhibit the mitochondrial electron-transport chain, causing a decrease in oxygen consumption.

The rate of oxygen consumption in *B. cinerea* was reduced by 46% when the fungus was treated with 2-alylphenol (640 mM), while treated with antimycin A, the rate was reduced by 77% [48]. On the other hand, it has been reported that 5,7-dihydroxy-3,8-dimethoxyflavone, a flavonoid isolated from resinous exudates of *Pseudognaphalium robustum*, also inhibits the oxygen consumption in *B. cinerea* by 35.4% [51]. The results of the oxygen consumption test and those carried out with SHAM suggest that at least compound 7 would cause an inhibitory effect in the electron-transport chain.

## 4. Conclusions

A series of 12 new carboxamide-type compounds were synthesized from natural sesquiterpene drimenol. All of these compounds inhibited the mycelial growth of *B. cinerea*. Ring substituents modified the fungitoxic effect of the compounds, and the most active compounds were 6, 7, 12, and 13. Compounds 6 and 7 showed a behavior similar to known inhibitors of the electron transport chain when they act in conjunction with SHAM.

## Figures and Tables

**Figure 1 jof-07-00902-f001:**
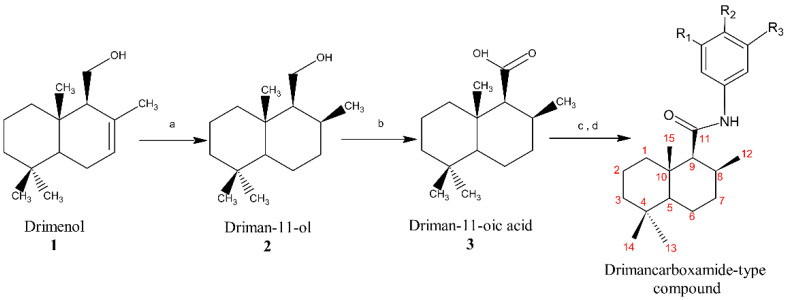
Route of synthesis of drimancarboxamide -type compounds. a. H_2_, Pd/C; b. acetone; CrO_3_, H_2_SO_4_; c. SOCl_2_, DMF; d. aniline, triethylamine, CH_2_Cl_2_.

**Figure 2 jof-07-00902-f002:**
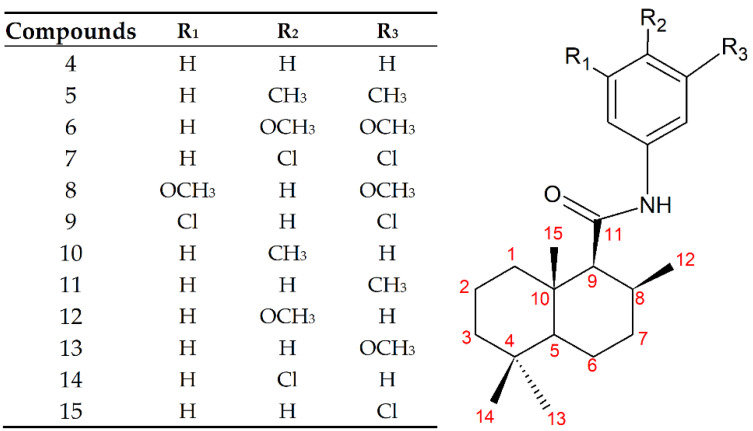
Synthesized compounds.

**Figure 3 jof-07-00902-f003:**
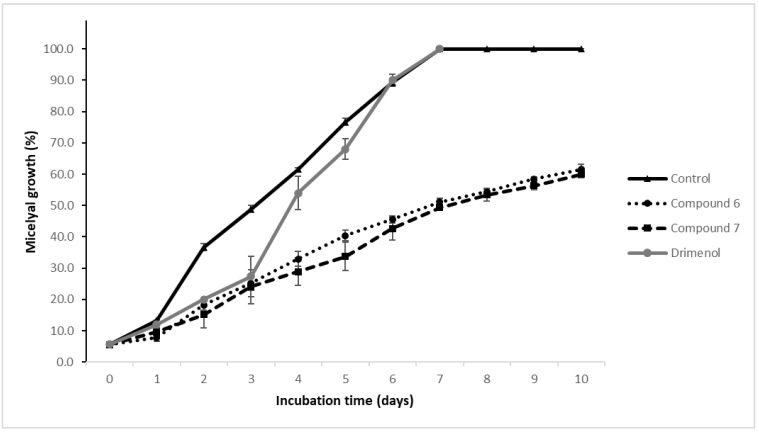
Effect of drimenol, compounds 6 and 7 (0.29 mM) on mycelial growth of *B. cinerea*. The data correspond to three independent assays in triplicate. The bars correspond to the standard deviation.

**Figure 4 jof-07-00902-f004:**
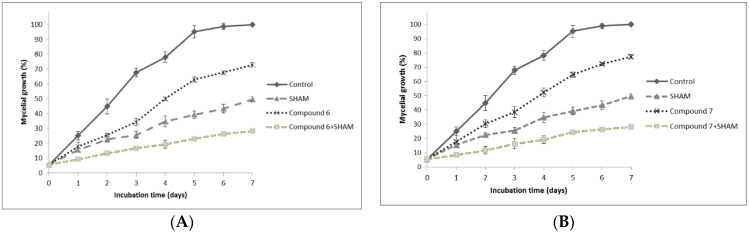
Effect of compound **6** (**A**) and compound **7** (**B**) at a concentration of 0.26 and 0.20 mM, respectively, on *B. cinerea* mycelial growth in the presence or absence of SHAM. Data shown correspond to three independent essays. Each essay was done in triplicate, and bars correspond to standard deviations.

**Figure 5 jof-07-00902-f005:**
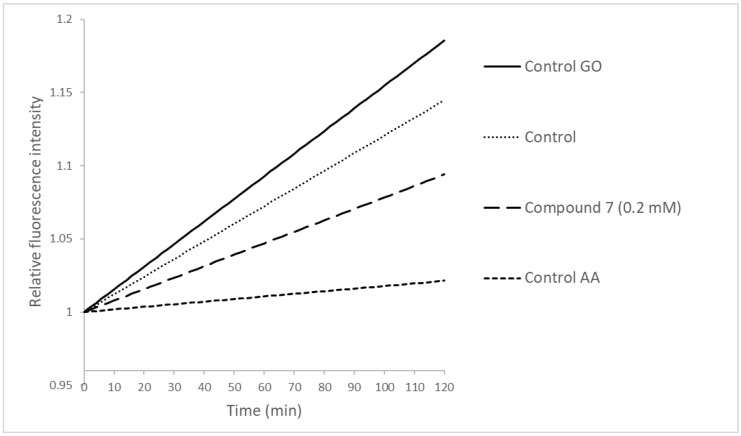
Effect of compound 7 on the oxygen consumption rate measured by MitoXpress fluorescence. “Control GO” corresponds to samples without conidia and glucose oxidase enzyme as the positive control. “Control AA” corresponds to conidia treated with antimycin A (a respiratory chain inhibitor). Control corresponds to conidia without any treatment. Two independent assays were done. Each assay was done in triplicate.

**Table 1 jof-07-00902-t001:** Fungitoxicity on *B. cinerea* of the synthesized compounds.

Compounds	IC_50_ (mmol/L)
drimenol	-
4	69.90 ± 4.60
5	5.18 ± 0.41
6	0.26 ± 0.04
7	0.20 ± 0.02
8	1.31 ± 0.13
9	8.17 ± 0.37
10	4.59 ± 0.32
11	6.04 ± 0.17
12	0.23 ± 0.03
13	0.26 ± 0.06
14	0.32 ± 0.03
15	3.53 ± 0.16

## Data Availability

Not applicable.

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
