# Peer review of "Characterization of the Fungitoxic Activity on Botrytis cinerea of N-phenyl-driman-9-carboxamides"

_jof, 2021, doi:10.3390/jof7110902_

Round 1

Reviewer 1 Report

The article “Characterization of the fungitoxic activity on Botrytis cinerea of N-phenyl-driman-9-carboxamides” by Mendoza and Cotoras et al., is focused to the study of the fungitoxic activity of 12 synthesized drimancarboxamides. Furthermore, the effect of these compounds on the mycelial growth and conidia germination of B. cinerea were studied.

This is an interesting and satisfactory paper, which will be of interest to the readerships and deserve be published in the journal of fungi.

However, the authors before publishing must clarify several small details in the text.

Title:

N-phenil-driman…... instead of................ n-phenyl…

Introduction:

Authors, reference 1 should be updated. Currently B. cinerea affects a total of 586 genera of vascular plants, representing over 1400 ornamental and agriculturally important plant species.

  • Fillinger, S., Elad, Y., 2016. Botrytis – the Fungus, the Pathogen and its Management in Agricultural Systems. Springer International Publishing Switzerland.
  • Elad, Y., Vivier, M., and Fillinger, S., 2016. Botrytis, the good, the bad and the ugly, in: Botrytis – the Fungus, the Pathogen and its Management in Agricultural Systems. Edit. Springer International Publishing Switzerland, pp 1-15

Experimental:

Page 2, line 80 drimenol…., and line 92.. driman-11-ol,  indicate the numbers (1) and (2)

The NMR spectroscopic constants for compound 1-3 have previously been reported. These data could be showed in the supporting information, and it should be deleted of the manuscript to save space.

Page 3, line 104

2.2.3 Reaction of halogenation and substitution.

Although, formally a halogen is substituted by an amine group, the reaction to obtain carboxamide is an addition-elimination reaction, no a substitution.

I suggest change this text header for,… 2.2.3 Synthesis of drimancarboxamide derivatives. General procedure.

The paragraph between lines 105-114 should be reworded, it is difficult to understand.

Results and discussion

Figure 1. Numbering some structure of drimanyl skeleton

Figure 1. Drimancarboxamide type compound instead of carboxamide type compound

In the text of figure 1, leave only   ….. a. (reagent), b. (reagent)……

Remove type of reaction, Hydrogenation, oxidation, halogenation and substitution

Line 288, Twelve new drimancarboxamides were……

Figure 2. Synthesized driman-9-carboxamides... instead of      synthesized compounds

Reviewer 2 Report

Dear Editor, in the manuscript the authors describe 12 compounds obtained by synthesis using drimenol as starting material.The compounds were then evaluated for their fungitoxic activity against B. cinerea. Overall, the paper is interesting but several points need to be clarified before its final acceptance. Please, see my comments below and,mainly, my first 5 comments that I consider the major one.

  • The NMR spectra for the compounds must be given as supplementary material. It seems to me that some H are missing from the peak list.
  • Line 109: how many mmol of TEA were added?
  • Are the authors sure that the stereochemistry of final compounds is retained in the synthetic procedure? Especially for C9.
  • Since the authors gave the absolute stereochemistry of the compounds, a chiral HPLC or other suitable technique should be used to measure the enantiomeric purity of the final compound.
  • Table 1: The IC50 value of the reference compound should be added.
  • Line 407: the sentence "the fruit of Ginkgo biloba.."is uncorrect. The plant is a gymnosperm, it has not an ovary and it can not produce a fruit. Please, revise it.
  • The synthetic scheme must be changed. The acyl choride was not isolated. Above the 3th arrow, they should put c, d and, therefore the structures of chloride and anilines must be removed
  • Line 3: please, change "n-phenyl-driman-9-carboxamides" to "N-phenyl-driman-9-carboxamides"
  • line 13: please, change "N-phenyl...." to "N-phenyl..."
  • line 20: please, change " (6) and one chlorinated compound (7)" to " (6) and one chlorinated compound (7) " (with numbering in bold)
  • line 52: I suggest to change "bark of cinnamon" to "bark of winter cinnamon"
  • lines 52 and 408: please, add the author of the botanical species
  • line 56: please, put the meaning of SDHI (since it is the mention in the text)
  • line 62: please, change "reactives" with "reagents"
  • line 73; please, use the symbol ° for Celsius degrees (there is a line below the symbol). Please, check the whole manuscript
  • line 83: please, specify the solvents ratio in the mobile phase
  • some sentences are not clear. Please, edit english language (see for example lines 92-96)

Round 2

Reviewer 2 Report

  • In table 1, the IC50 value of drimenol was not added. If the exact value is not available, because it is out of the tested range, the authors can write IC50>...
  • I have just noticed that the concentration values are expressed (at line 235) in µg/mL. I suggest to convert them in mmol/L
